# Therapeutic Potential of Targeting Ferroptosis in Periprosthetic Osteolysis Induced by Ultra-High-Molecular-Weight Polyethylene Wear Debris

**DOI:** 10.3390/biomedicines13010170

**Published:** 2025-01-13

**Authors:** Takuya Ogawa, Shunichi Yokota, Liyile Chen, Yuki Ogawa, Yoshio Nishida, Taiki Tokuhiro, Hend Alhasan, Tomoyo Yutani, Tomohiro Shimizu, Daisuke Takahashi, Takuji Miyazaki, Tsutomu Endo, Ken Kadoya, Mohamad Alaa Terkawi, Norimasa Iwasaki

**Affiliations:** 1Department of Orthopedic Surgery, Faculty of Medicine, Graduate School of Medicine, Hokkaido University, Kita-15, Nish-7, Kita-ku, Sapporo 060-8638, Japan; tkyoztf@yahoo.co.jp (T.O.); falcon8863@gmail.com (S.Y.); drclyl@gmail.com (L.C.); snow.stream828@gmail.com (Y.O.); nsd.yso@gmail.com (Y.N.); taikishuttle.t@gmail.com (T.T.); hendalhasan2011@gmail.com (H.A.); simitom@wg8.so-net.ne.jp (T.S.); rainbow-quest@pop02.odn.ne.jp (D.T.); takuzimiyazaki@gmail.com (T.M.); t.endo@med.hokudai.ac.jp (T.E.); kadoya@rf7.so-net.ne.jp (K.K.); niwasaki@med.hokudai.ac.jp (N.I.); 2R&D Center, Teijin Nakashima Medical Co., Ltd., Okayama 701-1221, Japan; t.yutani@teijin-nakashima.co.jp

**Keywords:** periprosthetic osteolysis, wear debris, ferroptosis inhibitors, therapeutics

## Abstract

**Background/Objectives:** Periprosthetic osteolysis is the primary cause of arthroplasty failure in the majority of patients. Mechanistically, wear debris released from the articulating surfaces of a prosthesis initiates local inflammation and several modes of regulated cell death programs, such as ferroptosis, which represents a promising therapeutic target in various chronic inflammatory diseases. Thus, the current study aimed at exploring the therapeutic potential of targeting ferroptosis in a polyethylene-wear-debris-induced osteolysis model. **Methods:** Inverted cell culture model was used for stimulating the cells with wear debris in vitro, and calvarial osteolysis model was used for evaluating the therapeutic effects of inhibitors in vivo. **Results:** The immunostaining of periprosthetic bone tissues demonstrated a number of osteocytes expressing ferroptosis markers. Likewise, the expressions of ferroptosis markers were confirmed in polyethylene-wear-debris-stimulated osteocyte-like cells and primary osteoblasts in a direct stimulation model but not in an indirect stimulation model. Furthermore, polyethylene wear debris was implanted onto calvarial bone and mice were treated with the ferroptosis inhibitors DFO and Fer-1. These treatments alleviated the inflammatory and pathological bone resorption induced by the wear debris implantation. **Conclusions:** Our data broaden the knowledge of the pathogenesis of periprosthetic osteolysis and highlight ferroptosis as a promising therapeutic target.

## 1. Introduction

As life expectancy increases, the number of patients receiving total joint arthroplasty is expected to continue rising worldwide. However, aseptic loosening due to periprosthetic osteolysis is the major complication leading to implant failure that requires urgent surgery. The surgical revision of TJA represents a major challenge to orthopedic surgeons and healthcare services, with a substantial economic burden worldwide [1]. Periprosthetic osteolysis is initiated when tissue macrophages recognize the wear debris released from articulating surfaces of prosthesis and promote low-grade chronic inflammation (LGI) in periprosthetic tissue. The persistence of the LGI state in tissue initiates local hypoxia and necrosis that further promote inflammation and local bone resorption, leading to implant loosening [2]. Therapeutic agents that inhibit inflammation and bone resorption have failed to prevent osteolysis and prolong the lifespan of implants. These facts underline the importance of research into developing effective therapy to manage this health problem.

The LGI state in periprosthetic tissue is known to promote osteoclast differentiation and induce several modes of regulated cell death (RCD), including apoptosis, pyroptosis, and ferroptosis in osteocytes and osteoblasts [3,4,5]. These changes disrupt the homeostatic function in the bone and cause a shift in the bone-remodeling process toward pathological bone resorption [2]. Growing evidence highlights that reactive oxygen species (ROS) are elevated in periprosthetic tissue and nearby bone tissues, and play a critical role in the pathogenesis of osteolysis [6]. The increased level of ROS amplifies mitochondrial lipid peroxidation, resulting in the development of the non-apoptotic cell death of ferroptosis [7]. Ferroptosis has been involved in a variety of pathological conditions in osteoblasts, as it disrupts their maturation/differentiation and function [8,9]. Therefore, ferroptosis has been implied in the pathogenesis of several musculoskeletal conditions, including osteoporosis, osteoarthritis, rheumatoid arthritis, and osteosarcoma. The major signaling pathways regulating ferroptosis include Xc-GSH-GPX4, FSP1-CoQ10-NAD(P)H, and GCH1-BH4 [5]. However, Xu and colleagues documented that the Nrf2-ARE signaling pathway regulates the osteoblast ferroptosis induced by CoCrMo nanoparticles [10].

Osteocytes are the most abundant cellular component in the mineralized bone matrix, accounting for 95% of the cells in bone. They are terminally differentiated cells of the osteoblast lineage, which plays a fundamental role in regulating bone remodeling through secreting cytokines [2]. They use the lacunar–canalicular system to communicate with other cells in bone and deliver their secretomes. The disruption of their function is associated with pathological changes in the bone microarchitecture in diseases, such as osteoarthritis and osteoporosis [11,12,13]. Importantly, the wear particles of UHMWPE, XLPE, Ti6Al4V, and CoCrMo are capable of stimulating osteocytes to produce inflammatory and osteoclastogenic cytokines that contribute to the progression of periprosthetic osteolysis [14,15]. In addition, there is evidence highlighting that both metal and polyethylene wear particles can induce caspase-dependent apoptosis and necrosis [15]. However, little is known about the involvement of ferroptotic osteocytes in the development of periprosthetic osteolysis.

In the current study, we investigated the ability of UHMWPE wear debris to activate ferroptosis pathway in MLO-Y4 osteocytes and the therapeutic potential of targeting ferroptosis in a wear-particle-induced osteolysis model.

## 2. Materials and Methods

### 2.1. Clinical Samples and Immunostaining

A bone sample was collected from a 55-year-old man who underwent a revision of a total hip arthroplasty. Samples were fixed by paraformaldehyde (Wako, Osaka, Japan), decalcified by treatment with EDTA (Wako, Osaka, Japan) for 2 months, and embedded in paraffin. The 3 μm sections were deparaffinized and subjected to antigen retrieval by treatment with proteinase K (Dako, Santa Clara, CA, USA) for 5 min. Next, the sections were first incubated with a blocking buffer containing 5% horse serum for 1 h and then with primary antibodies (1:200) to GPX4 (Abcam, Cambridge, UK) and NFE2l2 (NRF2, Gene Tex, Irvine, CA, USA), followed by treatment with a secondary antibody (EnVision+ System-HRP Labelled Polymer; Dako, Tokyo, Japan) for 30 min. The sections were subjected to Vectastain Elite ABC kit for the detection of horseradish peroxidase (HRP) (Vector Laboratories, Burlingame, CA, USA) and hematoxylin staining (Wako, Japan) for detecting cellular nuclei. The sections were examined by an All-In-One Fluorescence microscope (Keyence BZ-X710, Tokyo, Japan).

### 2.2. Preparation of Fabricated Wear of UHMWPE Debris

Virgin UHMWPE was manufactured from GUR1020 powder (Celanese Japan, Tokyo, Japan) exposed to 95 kGy irradiation (gamma radiation in a nitrogen atmosphere) and an annealing temperature of 135 °C or less. UHMWPE materials were crushed using a Multi Beads Shocker (Yasui Kikai, Osaka, Japan) at 3500 rpm, sterilized using ethylene oxide gas (EOG) sterilizer (Eogelk-SA-H160, Shiga, Japan), and placed in Eppendorf tubes [16]. The sizes were analyzed by a particle image analyzer Morphologi G3 (Malvern Instruments, Worcester, UK). The majority of particles had sizes of 0.1–0.99 µM (46.6%), 1–9.9 µM (33.4%), and >10 µM (20%).

### 2.3. Cell Isolation and Culture

Experimental procedures for mice were approved by the Institute of Animal Care and Use Committee of the Hokkaido University Graduate School of Medicine (approval IDs: 17-0085 and 24-0118). Murine osteoblasts were isolated from 7-day-old C57BL/6 mice (CLEA, Tokyo, Japan) calvarial bones digested by 25% trypsin solution (Wako, Osaka, Japan) and collagenase Type II (Sigma, Darmstadt, Germany) at 37 °C for 10 min with shaking. The osteoblasts were washed with sterile phosphate-buffered saline (PBS; Nacalai Tesque, Kyoto, Japan) and cultured in the growth medium that contained α-modification minimum essential medium (αMEM; Sigma) supplemented with 10% heat-inactivated fetal bovine serum, 1% penicillin/streptomycin solution (Wako), and 1% l-glutamine solution (Sigma). All experiments were performed using passage 2 of the isolated osteoblasts. For the macrophage isolation, 4% thioglycolate medium brewer Modified (Becton, Dickinson and Company BD, Sparks, MD, USA) were intraperitoneally injected in 8-week-old C57BL/6 mice (CLEA, Japan), and peritoneal macrophages were obtained from the peritoneal cavity after 5 days. The cells were washed with PBS (Nacalai Tesque) and cultured in the growth medium. MLO-Y4-like osteocytes (obtained from Professor Lynda F. Bonewald, Indiana Center for Musculoskeletal Health (ICMH, IU School of Medicine-Indianapolis, Indianapolis, IN, USA) were cultured in the same growth medium. In all the experiments, the cells were cultured in a 37 °C humidified atmosphere that contained 5% CO_2_.

### 2.4. Cell Stimulation and Viability Assays

For the direct stimulation of osteoblasts and MLO-Y4-like osteocytes with debris, the cells (1 × 10^5^) were seeded on coverslips and cultured with UHMWPE wear debris at 0.5 mg/mL for 48 h using an inverted method [16]. Briefly, the cells were seeded on a 15 mm coverslip (MATSUNAMI Glass, Osaka, Japan) and then carefully placed onto 200 µL growth medium with 0.5 mg/mL UHMWPE wear debris in a 12-well plate. For the indirect stimulation method, the macrophages (1 × 10^5^) were seeded on a 15 mm coverslip (MATSUNAMI) in a 12-well plate and cultured with UHMWPE wear debris at 0.5 mg/mL for 24 h using the inverted method. In parallel, the MLO-Y4/osteoblasts were placed (1 × 10^5^) in transwell inserts (1.0 μm Falcon cell culture inserts, BD, Sparks, MD, USA) and the macrophage culture was added after washing with PBS to remove the wear debris. The cells were co-cultured in a growth medium containing αMEM (Sigma) supplemented with 10% heat-inactivated fetal bovine serum, 1% penicillin/streptomycin solution (Wako), and 1% l-glutamine solution (Sigma) for 48 h. The viability analysis was performed on directly stimulated cells for 48 h incubation and assayed using Alamar Blue Cell Viability Reagent (Invitrogen, Winnipeg, MB, Canada). The results were determined using SpectaMax M5 (Molecular Devices, San Jose, CA, USA) at an excitation wavelength of 530 and emission of 590 nm.

### 2.5. Quantitative Real-Time Polymerase Chain Reaction

The cells were lysed via TRIzol Reagent (Invitrogen) and then subjected to an RNA extraction kit (Takara, Shiga, Japan). Purified 0.5 μg RNA samples were used to generate cDNAs using a GoScriptTM reverse transcriptase kit (Promega, Fitchburg, WI, USA). The cDNAs were then subjected to SYBR Premix Ex TaqTM II (Takara) and Thermal Cycler Dice (Real-Time System II, Takara, Japan). Reactions were prepared with specific primers to Gpx4 (forward: GGGGACGCTGCAGACAG, reverse: CTAGGACTTTGGCGTCCAGG), Nfe2l2 (forward: AATAAAGTCGCCGCCCAGAA, reverse: TGCTCCAGCTCGACAATGTT), Chac1 (forward: GCCCTGTGGATTTTCGGGTA, reverse: ATAGGCAAAGTCCGGCTTCC), and Gapdh (forward: TGCAGCGAACTTTATTGATG, reverse: ACTTTGTCAAGCTCATTTCC). Gene expression levels were calculated after normalization to the expression of the reference gene Gapdh via the 2^−ΔΔCt^ method.

### 2.6. Western Blot Analysis

The cultured cells were lysed using a RIPA lysis buffer (ATTO Corporation, Tokyo, Japan), and the extracted proteins were subjected to standard procedures for 12% SDS‒PAGE. Proteins were electrophoretically transferred onto polyvinylidene difluoride membrane (Immobilon-P Membrane; Merck, Darmstadt, Germany). Thereafter, the membranes were blocked in 5% skimmed milk and then incubated with each primary antibodies at a dilution of 1:1000, including β-actin (Bioss Antibodies, Beijing, China), GPX4 (Abcam), and NFE2l2 (Gene Tex). The blots were then incubated with the corresponding secondary HRP-conjugated antibodies (Cell Signaling Technology CST, Danvers, MA, USA) for 1 h, and the bands were visualized using an iBright 1500 Imaging System (Thermo Fisher Scientific, Inc., Waltham, MA, USA) after treatment with an EZ WestLumi Plus Kit (ATTO, Tokyo, Japan). Finally, the relative densities of detected bands were quantified using ImageJ software (National Institutes of Health NIH, Bethesda, MD, USA).

### 2.7. Calvarial Osteolysis Model and Lesions Examination

Debris-induced osteolysis was induced by the implantation of 6 mg fabricated wear UHMWPE debris onto the surface of the calvarial bone of 10-week-old C57BL/6 male mice for 7 days. The sagittal incision (~1 cm) was made over the calvarial anterior site after anesthetizing the mice with an intraperitoneal injection of 100 mg/kg ketamine and 10 mg/kg xylazine. Deferoxamine (DFO) (Cayman Chemical, Ann Arbor, MI, USA) 100 mg/kg, Ferrostatin-1(Fer-1) (Cayman Chemical, USA) 5 mg/kg, or Necrostatin-1 (Nec-1) (Cayman Chemical, USA) 3.2 mg/kg were intraperitoneally injected on days -1, 3, and 5 post-debris implantation. The pathological bone erosions were evaluated on day 7 using high-resolution micro-computed tomography assessment (micro-CT) and histopathology. The calvariae were subjected to micro-CT scanning R_mCT2 (Rigaku, Tokyo, Japan) at a 10 mm isotropic resolution, and the percentage of bone pits was calculated by ImageJ (NIH). For the histology, the calvariae were fixed in 10% formalin (Wako, Japan), decalcified in 10% EDTA (Wako, Osaka, Japan) for 3 days, and embedded in paraffin. The paraffin blocks were sectionized and stained with hematoxylin and eosin and tartrate resistance acid phosphatase (TRAP) staining (Wako, Japan). The microscopy images obtained by the All-In-One Fluorescence microscope (Keyence, Tokyo, Japan) were subjected to ImageJ (NIH) for the assessment of the pathological lesions and number of TRAP-positive cells [16].

### 2.8. Statistical Analysis

Statistical analyses were performed using GraphPad Software Prism 10.4.1 (San Diego, CA, USA; https://www.graphpad.com/). We used Student’s test for comparing the differences between two independent groups and the one-way analysis of variance (ANOVA) followed by Tukey’s multiple-comparison procedure for the comparison of the differences between different independent groups. The results were considered statistically significant when <0.05.

## 3. Results

### 3.1. Detection of Ferroptotic Osteocytes in Periprosthetic Bone Tissues

To gain evidence that ferroptosis occurs in osteocytes, periprosthetic bone tissues were obtained from patients undergoing revision surgery of a total hip arthroplasty and stained using specific antibodies to NRF2 and GPX4. Positive signals corresponding to NRF2 and GPX4 staining were detected in the osteocytes within lacunae (Figure 1). It is noteworthy that the number of NRF2-positive osteocytes was greater than that of the GPX4-positive osteocytes in the periprosthetic bone tissues. These results suggest that ferroptotic osteocytes were present in the periprosthetic bone tissues.

### 3.2. Expression of Ferroptosis Markers in UHMWPE-Wear-Debris-Simulated Osteocytes

To obtain additional evidence of the occurrence of ferroptosis in osteocytes, MLO-Y4-like osteocytes were cultured with UHMWPE wear debris and the expressions of ferroptosis markers were examined after 48 h (direct stimulation model). In parallel, peritoneal macrophages were stimulated with UHMWPE wear debris and then co-cultured with MLO-Y4-like osteocytes (indirect stimulation model). The gene expressions of ferroptosis markers in MLO-Y4-like osteocytes were examined after the 48 h culture. Primary osteoblasts were used as controls and tested in the two stimulation models for their gene expressions. Directly stimulated osteocytes with wear debris displayed a decreased expression of Gpx4 accompanied with an elevated level of Nfe2l2, which appeared comparable to those in directly stimulated osteoblasts (Figure 2A,B). On the other hand, co-cultured MLO-Y4-like osteocytes with debris-stimulated macrophages showed a significant increase in Gpx4 and decreases in Chac1 and Nfe2l2 (Figure 2C). Such changes were not found in the stimulated osteoblast in the same model (Figure 2D). The Western blot analysis demonstrated that the MLO-Y4-like osteocytes stimulated with UHMWPE wear debris but not macrophages exhibited a decrease in the expression of GPX4 with an increase in the expression of NRF2. These collective results indicate the activation of the ferroptosis pathway in periprosthetic bone tissues and highlight it as a therapeutic target.

#### Therapeutic Effects of Targeting Ferroptosis in Periprosthetic Osteolysis Model

To examine the beneficial effects of targeting ferroptosis in vivo, UHMWPE wear debris was implanted onto calvarial bone and mice were treated with DFO, Fer-1, or Nec1 on days −1, 3, and 5 post-debris implantation. The calvarial bone tissues were collected on day 7 for histomorphometric and histological evaluations. The micro-CT analysis revealed that the treatments with DFO and Fer-1, but not Nec1, alleviated the bone loss induced by the UHMWPE-wear-debris implantation (Figure 3A). The therapeutic effects of DFO and Fer-1 in suppressing the bone loss were comparable and the treated mice exhibited a minor bone osteolytic area that was the same as in the sham mice. Consistent with these findings, these mice exhibited a smaller osteolytic bone area with a reduction in the number of TRAP+ cells and inflammatory infiltrates, suggesting that the ferroptosis inhibitors effectively suppressed the pathological changes induced by the UHMWPE wear debris (Figure 3B). Our collective results highlight that ferroptosis inhibitors represent promising therapeutic agents for the prevention of periprosthetic osteolysis.

## 4. Discussion

A growing body of evidence that underlines a strong link between bone-related diseases and ferroptosis of osteocytes suggests that targeting this pathway offers a promising future therapeutic approach [17]. Little information regarding the involvement of ferroptotic osteocytes in the development of periprosthetic osteolysis and the usefulness of targeting this pathological process is available. Here, we demonstrated that UHMWPE wear debris promoted the expression of ferroptotic markers in osteocyte-like cells in vitro and treatment with ferroptosis inhibitors suppressed the development of osteolytic lesions in the osteolysis murine model.

The expressions of Gpx4, Chac1, and Nfe2l2, which are the well-accepted biomarkers of ferroptosis, were examined in stimulated osteocytes with UHMWPE wear debris. The stimulated cells displayed a decreased expression of Gpx4 in contrast to Nfe2l2, which indicates the impairment of the antioxidant response and the increased level of oxidative stress in the cells. In the osteolysis model, treatment with DFO or Fer1, but not Nec-1, alleviated the pathological lesions induced by the UHMWPE wear debris implantation. Our results appear consistent with earlier findings demonstrating that Fer1 treatment ameliorated the severity of osteolysis via reducing the osteoblast ferroptosis induced by CoCrMo nanoparticles [10]. Likewise, a blockade of oxidative stress reduced the osteolysis induced by titanium particles [18]. In line with these findings, targeting ferroptotic osteocytes protects from excessive bone loss in postmenopausal osteoporosis, along with other bone osteolytic diseases [12,13,19,20]. Fer1 and DFO treatments seem to have beneficial effects in several experimental diseased models, including ACLT-induced OA, thioacetamide-induced acute liver injury, and Parkinson’s disease models [21,22,23]. Fer1 is a synthetic antioxidant compound that acts as a hydroperoxyl radical scavenger in the presence of iron and prevents damage to membrane lipids and the consequent cell death. DFO is another ferroptosis inhibitor via binding free intracellular Fe3+, reducing ROS, and up-regulating the intracellular levels of GPX4 and the ferritin heavy chain (FTH1). In contrast, the Nect1 treatment was not effective at reducing the bone loss induced by the UHMWPE wear debris implantation, although this treatment was proved to inhibit osteoporosis progression via reducing necroptotic osteocytes in bone in ovariectomized rats [24]. Therefore, treatment with either Fer1 or DFO seemed to rescue osteocytes from death by regulating the increased lipid peroxides and ORS and restoring the function of GPX4 in the cells. These collective findings suggest that targeting ferroptosis rather than necroptosis is a promising therapeutic strategy for the prevention of periprosthetic osteolysis.

Ferroptosis is an intracellular iron-dependent form of cell death featuring an imbalance of the redox state with increased levels of intracellular reactive oxygen species (ROS) and is implicated in numerous pathological conditions, such as cancer, ischemic organ injuries, and neurodegenerative diseases [17]. The increased productions of lipid peroxidation initiate membrane instability and permeabilization, and eventually lead to cell death. Before cell death, ferroptotic cells display distinct morphologies from apoptosis, necrosis, and autophagy, with a characteristic round shape cells with no rupture in the plasma membrane and swelling in their cytoplasm or organelle [25]. GPX4 is known as the central regulator of ferroptosis through reducing phospholipid hydroperoxide, thereby protecting against excessive lipid peroxidation in the cells [26]. Its vital role in the maintenance of a variety of physiological functions is evident in specific genetic ablation mice that exhibit lipid-oxidation-induced acute renal failure with high mortality and the mitochondrial damage of neurons accompanied by hippocampal neurodegeneration [27,28]. Therefore, the approach to activate GPX4-inhibiting ferroptosis holds promise for the development of novel therapeutics [17]. Importantly, the ability of Fer-1 and DFO treatments to reverse the decrease in mRNA levels of GPX4 in the affected cells provide another possibility for their protective mechanism in various disease models [27,28].

Limitations of the current study include the following: (1) The experimental model applied UHMWPE wear debris to calvarial bone, not into a joint, which may not fully recapitulate the natural progression of periprosthetic osteolysis. (2) This study employed ferroptosis inhibitors that are not yet approved for clinical use. (3) The in vitro model utilized MLO-Y4-like osteocytes, not primary cells, which may show a different response in vitro. Despite these limitations, our findings provide preliminary evidence for the activation of ferroptosis pathways in osteocytes and highlight this pathway as a potential therapeutic target for periprosthetic osteolysis.

## 5. Conclusions

Our findings revealed that UHMWPE wear debris activates the ferroptosis pathway in osteocytes and osteoblasts, and targeting this pathway is a promising therapeutic approach for the prevention of periprosthetic osteolysis. While ferroptosis inhibitors are not currently applied in clinical treatment, this promising therapeutic target presents an excellent avenue for the development of future treatments that aim at extending the lifespan of orthopedic implants, and consequently, enhancing the quality of life for patients.

## Figures and Tables

**Figure 1 biomedicines-13-00170-f001:**
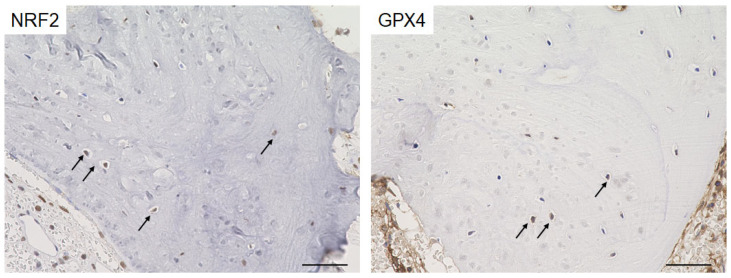
Detection of GPX4- and NRF2-positive osteocytes in periprosthetic bone tissues. IHC staining of the bone tissues from patients who underwent revision surgery with specific antibodies. Arrows indicate positive reactions. Scale bars are 50 µm.

**Figure 2 biomedicines-13-00170-f002:**
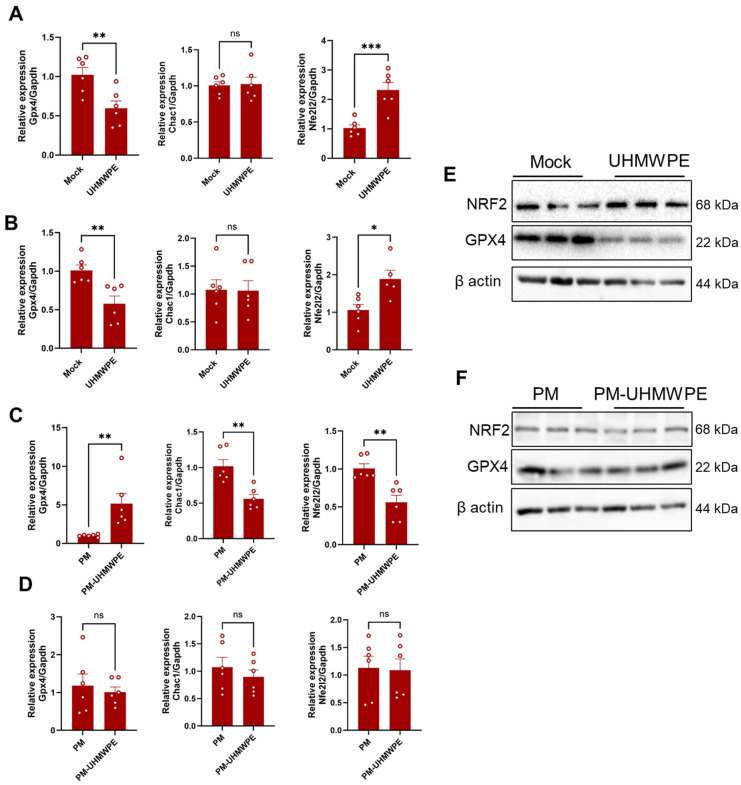
Expressions of ferroptosis markers in UHMWPE-wear-debris-simulated osteocytes and osteoblasts. Gene expressions of Gpx4, Chac1, and Nfe2l2. (**A**,**B**) Relative gene expressions of stimulated MLO-Y4-like osteocytes (**A**) and osteoblasts (**B**) in the direct stimulation model with UHMWPE debris. (**C**,**D**) Relative gene expressions of stimulated MLO-Y4-like osteocytes (**C**) and osteoblasts (**D**) in an indirect stimulation model using the UHMWPE-debris-stimulated macrophages. The results represent the means of 6 samples ± SEM. The significant difference between the two groups was determined by a two-tailed Student’s *t*-test. (**E**,**F**) Protein expression analysis of NRF2 and GPX4 in the stimulated MLO-Y4-like osteocytes. (**E**) Direct stimulation model. (**F**) Indirect stimulation model. * = *p* < 0.05; ** = *p* < 0.01; *** = *p* < 0.001; ns = not significant.

**Figure 3 biomedicines-13-00170-f003:**
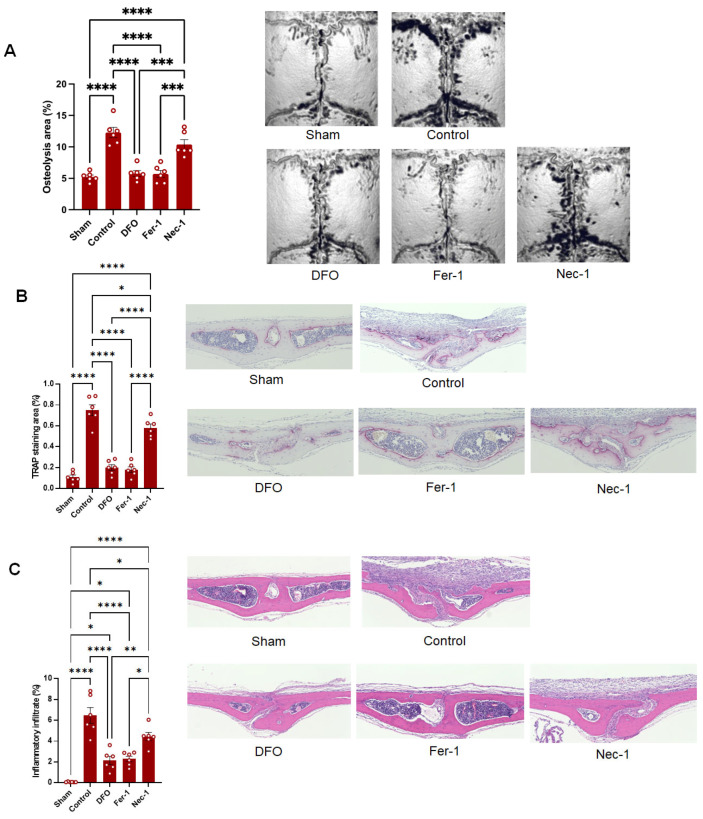
Evaluation of the beneficial effects of targeting ferroptosis in the inflammatory osteolysis model. The sham mice did not receive the debris implantation, while all the other groups received it. (**A**) Quantification of the bone resorption area in the calvarial bone tissues analyzed by micro-CT. The right panels are representative images for the micro-CT. (**B**) Quantification of the TRAP-stained areas in the calvarial bone sections. The right panels are representative images for the TRAP-stained sections. (**C**) Quantification of inflammatory infiltrates in the calvarial bone sections. The right panels are representative images for the HE-stained sections. The results represent the means ± SEM of 6 mice. Scale bars are 50 µm. The significant difference between the two groups was determined by one-way ANOVA, followed by Tukey’s multiple-comparison procedure. * = *p* < 0.05; ** = *p* < 0.01; *** = *p* < 0.001; **** = *p* < 0.0001.

## Data Availability

The data in this study are available within this article and other data will be made available upon reasonable request from the corresponding author.

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
