# Peer review of "Therapeutic Potential of Targeting Ferroptosis in Periprosthetic Osteolysis Induced by Ultra-High-Molecular-Weight Polyethylene Wear Debris"

_biomedicines, 2025, doi:10.3390/biomedicines13010170_

Round 1
Reviewer 1 Report
Comments and Suggestions for Authors
This manuscript represents one stage of a global study aimed at developing effective therapeutic agents for the prevention of periprosthetic osteolysis. Authors have at least managed to find one possible solution to this important and urgent problem.
The "Introduction" section is presented briefly, but quite informatively to justify the aim and tasks of the research. The literary sources used for the review are relevant. Other sections of the manuscript are also presented at a sufficiently high level, understandable to specialists in this field of research.
In my opinion, the manuscript can be recommended for publication without any changes or additions.
Author Response
Thank you very much for kind words and the outstanding performance review as well as appraisal.
Reviewer 2 Report
Comments and Suggestions for Authors
The topic of this paper is interesting and might be important in clinical medicine. The paper is in general well-written; the introduction provides broad overview of the field and rationale for conducting this study. Conducted methods are described in detail and obtained results are properly discusses.
However, there are some issues regarding the presentation and discussion of results that should be addressed prior publication.
1. Figure 2 should be reorganized to improve the visibility of individual parts. In sections A-D, the letters are very small, making it difficult to read the results.
2. I would strongly suggest to expand the textual description of the results (especially in the section 3.3. Therapeutic effects of targeting ferroptosis in periprosthetic osteolysis model)
3. I would suggest to better describe clinical implication and future perspectives of findings of this study.
Author Response
We are grateful to the reviewer for the constructive comments and suggestions that helped us to significantly improve our manuscript. We have revised the manuscript according to your comments and suggestions.
Comment 1
Figure 2 should be reorganized to improve the visibility of individual parts. In sections A-D, the letters are very small, making it difficult to read the results.
We acknowledge the reviewer’s suggestion and we have reorganized images for better clarity and visibility.
Comment 2.
I would strongly suggest to expand the textual description of the results (especially in the section 3.3. Therapeutic effects of targeting ferroptosis in periprosthetic osteolysis model)
We acknowledge the reviewer’s suggestion and we have added a new paragraph for describing the results of osteolysis model. The new paragraph is as follows: “To examine the beneficial effects of targeting ferroptosis in vivo, UHMWPE wear debris were implanted onto calvarial bone and mice were treated with DFO, Fer-1 or Nec1 on days -1, 3, and 5 post debris implantation. Calvarial bone tissues were collected on day 7 for histomorphometric and histological evaluations. The micro-CT analysis revealed that treatment with DFO and Fer-1, but not Nec1 alleviated bone loss induced by UHMWPE wear debris implantation (Fig. 3A). Therapeutic effects of DFO and Fer-1 in suppressing the bone loss were comparable and treated mice exhibited minor bone osteolytic area as same as sham mice. In consistent with the findings, these mice exhibited smaller osteolytic bone area with a reduction in the number of TRAP+ cells and inflammatory infiltrates, suggesting that ferroptosis inhibitors effectively suppressed the pathological changes induced by UHMWPE wear debris (Fig. 3B).
Comment 3.
I would suggest to better describe clinical implication and future perspectives of findings of this study.
We acknowledge the reviewer’s suggestion, and we have added a new paragraph for describing the clinical implication and future perspectives in the conclusion section as follows: “While ferroptosis inhibitors are not currently applied in clinical uses, this promising therapeutic target presents an excellent avenue for the development of future treatments that aim at extending the lifespan of orthopedic implants and consequently enhancing the quality of life for patients.”